# Health-Screening-Based Chronic Obstructive Pulmonary Disease and Its Effect on Cardiovascular Disease Risk

**DOI:** 10.3390/jcm11113181

**Published:** 2022-06-02

**Authors:** Sang-Jun Lee, Sung-Soo Yoon, Myeong-Hoon Lee, Hye-Jun Kim, Yohwan Lim, Hyewon Park, Sun Jae Park, Seogsong Jeong, Hyun-Wook Han

**Affiliations:** 1Department of Biomedical Informatics, School of Medicine, CHA University, Seongnam 13488, Korea; sangjun9716@gmail.com (S.-J.L.); thanks9157@gmail.com (S.-S.Y.); lmh164231@gmail.com (M.-H.L.); kimhj95@yonsei.ac.kr (H.-J.K.); dladyghks918@gmail.com (Y.L.); peacock1128@naver.com (H.P.); 2Institute of Basic Medical Sciences, School of Medicine, CHA University, Seongnam 13488, Korea; 3Institute for Biomedical Informatics, School of Medicinie, CHA University, Seongnam 13488, Korea; 4Department of Biomedical Sciences, Seoul National University College of Medicine, Seoul 03080, Korea; sunjaepark.a@gmail.com; 5Healthcare Big-Data Center, Bundang CHA Hospital, Seongnam 13488, Korea

**Keywords:** chronic obstructive pulmonary disease, cardiovascular disease, score system, self-diagnosis

## Abstract

Chronic obstructive pulmonary disease (COPD) is considered a major cause of death worldwide, and various studies have been conducted for its early diagnosis. Our work developed a scoring system by predicting and validating COPD and performed predictive model implementations. Participants who underwent a health screening between 2017 and 2020 were extracted from the Korea National Health and Nutrition Examination Survey (KNHANES) database. COPD individuals were defined as aged 40 years or older with prebronchodilator forced expiratory volume in 1 s/forced vital capacity (FEV1/FVC < 0.7). The logistic regression model was performed, and the C-index was used for variable selection. Receiver operating characteristic (ROC) curves with area under the curve (AUC) values were generated for evaluation. Age, sex, waist circumference and diastolic blood pressure were used to predict COPD and to develop a COPD score based on a multivariable model. A simplified model for COPD was validated with an AUC value of 0.780 from the ROC curves. In addition, we evaluated the association of the derived score with cardiovascular disease (CVD). COPD scores showed significant performance in COPD prediction. The developed score also showed a good effect on the diagnostic ability for CVD risk. In the future, studies comparing the diagnostic accuracy of the derived scores with standard diagnostic tests are needed.

## 1. Introduction

Chronic obstructive pulmonary disease (COPD) is the third leading cause of death in the world, and its global burden is expected to further increase [1]. Several studies suggest that comorbid conditions such as cardiovascular disease (CVD) and poor health-related quality of life influence the worsening of respiratory symptoms in COPD patients [2]. The extrapulmonary effect showing symptoms of skeletal muscle dysfunction and osteoporosis is also reported to be frequent in COPD patients with some diseases, such as chronic infections and CVD [3]. COPD exacerbation is important because it accelerates the progression of other closely related diseases, and if prevented, it can improve the health-related quality of life, prolong life and reduce health care costs [4,5,6].

As noted in the 2006 update of the Global Initiative for Obstructive Lung Disease (GOLD) guidelines, the definition of COPD is a preventive and treatable disease [7]. However, it is difficult to find out whether an individual has COPD due to low hospital visit rates, lack of knowledge about symptoms and disease, and the complexity of the COPD diagnosis method [8]. According to a recent nationwide survey conducted in Taiwan, the incidence of COPD is expected to be about 6 percent, but less than half of those may have COPD have undergone a spirometry test which is one of the most non-invasive tests used to diagnose COPD [9]. Regarding COPD diagnostic and definition issues, the GOLD recommended that COPD management and treatment should be based not only on spirometric findings but also on scores, such as the Chronic Respiratory Questionnaire [10,11]. Another well-known scoring system is the COPD Assessment Test (CAT), a scoring method using eight preliminary symptoms associated with COPD [12]. Although it is a sophisticated scoring system on a multidimensional scale, there is the limitation whereby it is required that the subject has individual awareness of the preliminary symptoms of COPD used in scoring, which are difficult to understand alone. With these issues, several previous prediction models for COPD were based on a combination of information from medical history, clinical characteristics and laboratory biomarkers [13]. Likewise, it has limitations due to the problem of clinical application, because the used prediction method is unclear, not validated or limited due to the diverse variables used for prediction.

Herein, we conducted this study to develop a simple COPD score directly applicable to the general population who undergo health screenings, which was validated in an independent cohort. In addition, the association of the developed score with CVD, including coronary heart disease (CHD) and stroke, was evaluated using the Korean National Health and Nutrition Examination Survey (KNHANES) database.

## 2. Materials and Methods

### 2.1. Study Population

KNHANES, conducted by Korea Disease Control and Prevention Agency (KCDC), is the representative national cross-sectional surveillance system that provides data for the evaluation of nutritional status, health policy effectiveness and trends in the prevalence of health risk factors and major chronic diseases [14]. The sampling population of each survey year consists of approximately 10,000 people. The database consists of three components: a health interview, a health examination and a nutrition survey. The health interview is conducted during an interviewer’s home visit. The health examinations and questionnaires include socioeconomic status, health behavior, quality of life, medical use, biochemical profile using fasting serum and urine, dental health, visual acuity, hearing, bone density measurements and X-ray test results. It also collects detailed information about food intake and eating habits. This study analyzed participants who underwent a health screening between 2017 and 2020 using the KNHANES database.

Figure 1 depicts the flow diagram for the inclusion of the study population after exclusion of participants aged below 20 and with missing information. We selected 7037 individuals in 2017–2018 for the training set and 3674 individuals in 2019–2020 for the validation set, respectively. This study was conducted in accordance with the Transparent Reporting of a Multivariable Prediction Model for Individual Prognosis or Diagnosis (TRIPOD) guidelines (Appendix A) [15]. The Institutional Review Board of CHA Bundang Hospital approved this study (No. 2022 04 041). Informed consents were waived because the database was provided for research purposes in an anonymized form under strict confidentiality guidelines.

### 2.2. Definition of Variables

COPD individuals were defined as aged 20 years or older with prebronchodilator forced expiratory volume in 1 s/forced vital capacity (FEV1/FVC < 0.7) [16]. Drinkers were defined as participants who drank at least once a week. Physical activity was defined as at least 150 min of moderate-intensity aerobic physical activity or 75 min of vigorous-intensity aerobic physical activity per week according to the modified Global Physical Activity Questionnaire [17]. CVD was defined as doctor-diagnosed CHD or stroke.

### 2.3. Statistical Analysis

All statistical analyses were performed using SAS version 9.4 (SAS Institute Inc., Cary, NC, USA). Continuous variables and categorical variables were presented as means (standard deviation (SD)) and numbers (%), respectively. The *t*-test was used for the continuous analyses, and the chi-squared test was used for the categorical analyses. Univariable and multivariable analyses were performed using the logistic regression model, which included odds ratios (ORs), 95% confidence intervals (CIs) and the concordance index (C-index). We used the purposeful selection used by Zhang et al. [18]. The variable selection with univariable analysis for the multivariable regression model was based on a significant level of *p* < 0.001. If more than one variable with a significant level of *p* < 0.001 was considered related, only variables with a higher C-index were included in the derivation after using the partial likelihood ratio test. For example, a multivariable model 1 with waist circumference (WC) and body mass index (BMI), and a multivariable model 2 with only WC are not significantly different in their fits for data. We chose model 2 for the principal of parsimony. After the univariable analyses, variables that required a hospital visit to be obtained were excluded for simplification and generalization. The following variables were selected for multivariable analyses: age (continuous; years), sex (categorical; men and women), WC (continuous; cm) and diastolic blood pressure (DBP; continuous; mmHg). The derived COPD score was validated in independent participants who underwent a health screening between 2019–2020. A receiver operating characteristic (ROC) curve with an area under the curve (AUC) value was performed using R version 4.1 (R Foundation for Statistical Computing, Vienna, Austria) and generated for the evaluation of sensitivity (Sens), specificity (Spec), positive predictive value (PV+) and negative predictive value (PV−). In addition, the validation cohort was stratified according to the interquartile range of the derived COPD score to determine the score-dependent ORs. Moreover, unadjusted ORs were calculated using logistic regression to confirm whether the derived score was informative in the stratification of individuals at higher risk of CVD, CHD and stroke.

## 3. Results

The numbers of participants with COPD and non-COPD at baseline were 949 and 6088, respectively (Table 1). Weight, triglyceride (TG), aspartate aminotransferase (AST), urinary glucose, urinary pH, smoking status and physical activity were not significantly different between COPD and non-COPD groups. Compared with non-COPD individuals, those with COPD were older men with higher systolic blood pressure (SBP), fasting serum glucose (FSG), blood urea nitrogen (BUN) and creatinine levels, but there was a lower proportion of smoking and physical activity. The descriptive characteristics of men and women are shown in Appendix A.

Nineteen variables from demographic characteristics, measurement, habits, blood pressure and tests for diabetes mellitus, dyslipidemia, liver function, kidney function and urine were evaluated in the univariable analyses (Table 2). The results indicate that the involved factors were generally and significantly reflective of COPD, except for weight (*p* = 0.508), TG (*p* = 0.911) and AST (*p* = 0.176). For covariate selection, the tests for diabetes mellitus, dyslipidemia, liver function, kidney function and urine were excluded in the simplification process, since these variables require a hospital visit. Subsequently, age, sex, BMI, WC, smoking, SBP and DBP remained potential candidates for the development of the COPD score. Between BMI and WC, WC had a higher C-index. As for SBP and DBP, DBP had a higher C-index. Finally, age, sex, WC and DBP were selected as components for the COPD score. In addition, the univariable analyses of variables among men and women are shown in Appendix A, respectively.

Table 3 shows the multivariable analysis of significant independent predictors selected in the univariable analyses. The multivariable model included age (OR, 1.087; 95% CI, 1.075–1.095; *p* < 0.001), sex (female; OR, 0.195; 95% CI, 0.164–0.230; *p* < 0.001), WC (OR, 0.981; 95% CI, 0.972–0.990; *p* < 0.001) and DBP (OR, 0.985; 95% CI, 0.977–0.993; *p* < 0.001), which constituted the KNHANES COPD score (C-index, 0.802). When smoking was further added as a component of the model, the difference in the C-index only increased by 0.01 (C-index, 0.812; Appendix A). Furthermore, the respective models developed in the male and female populations showed lower C-indices (0.773 for male, Appendix A; 0.756 for female; Appendix A).

According to the intercept and estimate values, the COPD score can be calculated as shown below. 

This is the equation for men:−3.582 + 0.083 × age − 0.019 × WC − DBP × 0.015(1)

This is the equation for women:−3.582 + 0.083 × age − 0.019 × WC − DBP × 0.015 − 1.636(2)

We then validated the derived score in the independent validation cohort, which revealed satisfactory accuracy (Figure 2). In Figure 2A, the AUC for the non-CVD validation cohort with no smoking covariate was 0.784, obtaining a Sens of 67.7%, a Spec of 76.5%, a PV+ of 6.2% and a PV− of 69.1%. In Figure 2B, the AUC for the non-CVD validation cohort with smoking covariate was 0.798, obtaining a Sens of 64.3%, a Spec of 81.9%, a PV+ of 6.4% and a PV− of 64.5%. In Figure 2C, the AUC for the validation cohort including CVD with no smoking covariate was 0.782, obtaining a Sens of 67.6%, a Spec of 76.2%, a PV+ of 6.3% and a PV− of 68.9%. In Figure 2D, the AUC for the validation cohort including CVD with smoking covariate was 0.795, obtaining a Sens of 70.2%, a Spec of 75.9%, a PV+ of 5.9% and a PV− of 68.4%.

After stratifying the participants according to the quartiles of the developed score, the 2nd (OR, 1.838; 95% CI, 1.156–2.923; *p* = 0.010), 3rd (OR, 2.128; 95% CI, 2.827–6.534; *p* < 0.001) and 4th (OR, 15.553; 95% CI, 10.484–23.071; *p* < 0.001) quartile groups were directly proportional in the validation dataset (Table 4).

When evaluating the association of the derived score with CVD risk to testify its applicability in the prediction of CVD, the KNHANES COPD score significantly estimated the risk of CVD (OR, 1.945; 95% CI, 1.776–2.130; *p* < 0.001), CHD (OR, 2.128; 95% CI, 1.896–2.390; *p* < 0.001) and stroke (OR, 1.661; 95% CI, 1.459–1.890; *p* < 0.001) in the training dataset (Table 5).

## 4. Discussion

We conducted this study to develop a self-diagnosis tool for COPD by simplification of the multivariable logistic model that exempted variables requiring hospital visits. The KNHANES COPD score consists of age, sex, WC and DBP, all of which are self-evaluable. The prediction score was developed with a C-index of 0.802 and validated with an AUC of 0.784 for the validation cohort including CVD. In addition, we developed a gender-stratified multivariable logistic model, a separate model due to the difference in COPD incidence and smoking rates between men and women. Through univariable analyses, it was confirmed that smoking was significantly different in the male group, and models when smoking was added and when smoking was not added are here presented. The developed COPD score was in direct proportion to the OR for COPD. Moreover, significant estimates were made for CVD risks using the COPD score. The effect of the developed COPD prediction score on CVD occurrence prediction was confirmed, and COPD prediction was possible with only the results of a general examination report. This suggests that there is potential to predict overall health, such as CVD and CHD.

COPD is a disease that exhibits airflow restriction due to airway resistance induced by the destruction of alveolar attachments as a result of the destruction of the pulmonary system and emphysema [19]. These pathological changes are caused by chronic inflammation of the periphery of the lungs; obstruction of the small airways before the occurrence of emphysema occurs first and gradually increases as the disease progresses [20,21]. The COPD severity stages are classified into five stages based on the FEV1-specific cut-point according to the GOLD guidelines, and as the stage increases, the inflammatory response is amplified through innate immune inflammatory response, airway remodeling and adaptive immune response [10,20]. When harmful inhalants, such as smoking, enter the airways, the first reaction is innate immunity. When harmful factors invade the airway, damage to lung epithelial cells is caused by recognition of TLR4 or TLR2 through innate immunity and the activation of NFkB, and epithelial cells produce and secrete many inflammatory mediators [22]. These inflammatory agents activate alveolar macrophages and neutrophils, and when proteases are released from them, they cause lung damage along with reactive oxygen specifications [23]. In addition, damage to epithelial cells, vascular endothelial cells and extracellular matrix, which are necrotized or self-destructed by lung damage, leads to many autoantigens, which the adaptive immune system recognizes as external antigens and causes an immune response [24]. Chronic immune-inflammatory reactions in these repeatedly damaged lung tissues lead to tissue repair and airway remodeling, which ultimately leads to airflow limitations such as pulmonary fibrosis [25]. Airflow restrictions can be seen as decreased FEV1/FVC and increased airway resistance and lung compliance [26].

COPD can increase the risk of other comorbidities. Severe airway obstruction has been reported to show a higher correlation [2], and it is claimed that inflammation in the lungs overflows into a systemic pattern [27]. In particular, a typical symptom of airway obstruction is gas exchange disorders such as pulmonary perfusion–ventilation imbalance, which causes hypoxemia, and in patients with actual emphysema, increased pulmonary vascular resistance and decreased alveolar area are also commonly observed [28]. Increased pulmonary vascular resistance in patients with COPD has been reported to play an important role in the development of pulmonary hypertension, one of the CVD risk factors [29], which increases the production and secretion of endothelin, a vasoconstrictor [30], and causes the smoothness of the channel [31]. Similarly, COPD is known to share risk factors such as coronary artery disease and age in old age, including smoking and lack of exercise [32]. Specifically, COPD patients are known to contribute to arteriovenous wall hardening due to decreased nitric oxide production of vascular endothelial cells and are considered risk factors for systemic hypertension and CVD [33].

The strength of this study is that it is the first study to develop and verify a COPD score system using only basic variables that do not require hospital visits. There are previous studies of various approaches using multivariable models or machine learning models for outcome exacerbation prediction. Of the 25 studies using the regression model or cox regression model, most of the models had overall lower performance than this study, and higher performance models also used predictors such as IgG titter and Gold stage, which are difficult to understand alone [13,34]. Clinical-data-based machine learning prediction models showed superior performance over an AUC of 0.80, but the variables used for prediction used 100-300 clinical features [35]. A machine learning model using data containing self-reports similar to this study also showed good performance, but the cat score, which is difficult to know immediately, was used [36]. Unlike previous studies, including systematic reviews related to COPD prediction using machine learning and statistical models, a practical self-diagnosis prediction model was here developed through the simplification of variables used for prediction and external verification. In addition, it showed good predictive performance for the diagnosis of COPD and the estimation of CVD risks. Taken together, the derived COPD score may support public health issues regarding socioeconomic costs and the application of non-face-to-face diagnosis for COPD [37].

This study includes several limitations that need to be considered. First, since this study is a cross-sectional and questionnaire-based study, it is difficult to identify causal relationships. Second, the study population consisted of the Korean population only. It is worth evaluating our model’s generalizability to other health care systems in other regions. Third, the golden criterion for detecting COPD is based on post-bronchodilator spirometry, and other diseases related to airflow disorders, such as bronchodilation and tuberculosis destruction, are likely to be included because FEV1/FVC <0.7 was used without utilizing chest radiography [7]. Finally, this model does not include biomarkers, such as blood eosinophils and fibrinogen, which are known to predict deterioration or hospitalization due to COPD, which should be the focus of future research.

## 5. Conclusions

In conclusion, the KNHANES COPD score, composed of age, sex, WC and DBP, satisfactorily predicted COPD. The developed score may be supportive in the stratification of individuals at high risk of COPD who require further screening for pulmonary diseases. Future studies comparing the diagnostic accuracy of the derived score with standard diagnostic tests are necessary to validate its accuracy and cost-effectiveness.

## Figures and Tables

**Figure 1 jcm-11-03181-f001:**
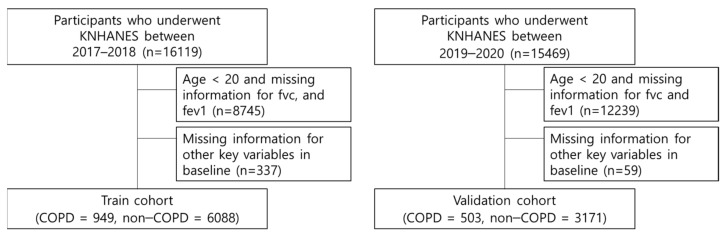
Flow diagram for the inclusion of the study population.

**Figure 2 jcm-11-03181-f002:**
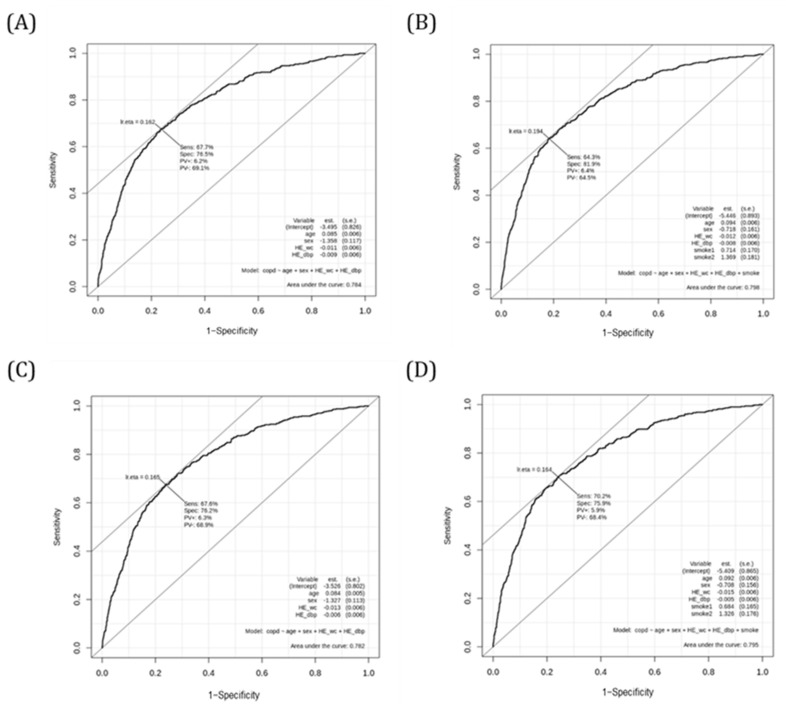
Receiver operating characteristic curves evaluating the performance of the COPD score for COPD in the validation cohort. (**A**) Receiver operating characteristic curve for non-CVD validation cohort with no smoking. (**B**) Receiver operating characteristic curve for non-CVD validation cohort with smoking. (**C**) Receiver operating characteristic curve for validation cohort including CVD with no smoking. (**D**) Receiver operating characteristic curve for validation cohort including CVD with smoking.

**Table 1 jcm-11-03181-t001:** Descriptive characteristics of the participants in the training cohort.

Characteristic	Overall (n = 7037)	Non-COPD (n = 6088)	COPD (n = 949)	*p*-Value
Age, years	58.4 (11.2)	57.1 (10.9)	66.6 (9.7)	<0.001
Sex, female, n (%)	3909 (55.5)	3647 (59.9)	262 (27.6)	<0.001
Height, cm	161.9 (8.8)	161.6 (8.7)	164.2 (8.8)	<0.001
Weight, kg	63.6 (11.0)	63.6 (11.1)	63.9 (10.3)	0.482
Body mass index, kg/m^2^	24.2 (3.2)	24.3 (3.2)	23.6 (2.8)	<0.001
Waist circumference, cm	83.3 (9.2)	83.1 (9.3)	84.8 (8.5)	<0.001
Systolic blood pressure, mmHg	121.2 (16.7)	120.8 (16.8)	123.7 (15.9)	<0.001
Diastolic blood pressure, mmHg	76.2 (10.1)	76.6 (10.0)	73.3 (10.2)	<0.001
Fasting serum glucose, mg/dL	104.0 (24.5)	103.7 (24.2)	106.1 (26.0)	0.006
Total cholesterol, mg/dL	195.2 (39.1)	196.7 (39.0)	185.9 (38.0)	<0.001
Triglyceride, mg/dL	141.2 (106.4)	141.2 (107.8)	141.6 (97.4)	0.905
Aspartate aminotransferase, IU/L	24.0 (11.5)	23.9 (11.8)	24.5 (9.1)	0.100
Alanine aminotransferase, IU/L	22.6 (14.9)	22.8 (15.3)	21.5 (11.6)	0.002
Blood urea nitrogen, mg/dL	15.6 (4.6)	15.5 (4.5)	16.7 (4.8)	<0.001
Creatinine, mg/dL	0.8 (0.2)	0.8 (0.2)	0.9 (0.3)	<0.001
Urinary protein				
Negative ^1^	6085 (86.2)	5271 (86.6)	794 (83.7)	0.082
Positive ^2^	972 (13.8)	817 (13.4)	155 (16.3)	
Urinary glucose				
Negative ^1^	6671 (86.2)	5782 (95.0)	889 (93.7)	0.056
Positive ^2^	972 (13.8)	306 (5.0)	60 (6.3)	
Urinary pH	5.9 (0.8)	5.9 (0.8)	5.9 (0.8)	0.585
Alcohol consumption, n (%)				
No	4012 (57.0)	3514 (57.7)	498 (52.5)	0.003
Yes	3025 (43.0)	2574 (42.3)	451 (47.5)	
Smoking, n (%)				
Never	4131 (58.7)	3821 (62.8)	310 (32.7)	0.633
Past	1722 (24.5)	1353 (22.2)	369 (38.9)	
Current	1184 (16.8)	914 (15.0)	270 (28.5)	
Physical activity, n (%)				
No	4347 (61.8)	3731 (61.3)	616 (64.9)	0.036
Yes	2690 (38.2)	2357 (38.7)	333 (35.1)	
Cardiovascular disease				
No	6686 (95.0)	5823 (95.6)	863 (90.9)	<0.001
Yes	351 (5.0)	265 (4.4)	86 (9.1)	
Chronic heart disease				
No	6820 (96.9)	5923 (97.3)	897 (94.5)	<0.001
Yes	217 (3.1)	165 (2.7)	52 (5.5)	
Stroke				
No	6883 (97.8)	5975 (98.1)	908 (95.7)	<0.001
Yes	154 (2.2)	113 (1.9)	41 (4.3)	

Data are mean (standard deviation) unless indicated otherwise. *p*-values calculated using *t*-tests (continuous variable) and chi-squared tests (categorical variable). Acronyms: COPD, chronic obstructive pulmonary disease. ^1^ negative. ^2^ positive (±, +, ++, +++, ++++).

**Table 2 jcm-11-03181-t002:** Univariable analyses of variables involved in the health examination for COPD in the training cohort.

Variable	Estimate	OR (95% CI)	*p*-Value	C-Index
Age, years	0.081	1.080 (1.080–1.090)	<0.001	0.743
Sex, female	−1.365	0.260 (0.220–0.300)	<0.001	0.661
Height, cm	0.034	1.030 (1.030–1.040)	0.011	0.592
Weight, kg	0.002	1.002 (0.996–1.008)	0.508	0.515
Body mass index, kg/m^2^	−0.067	0.940 (0.910–0.960)	<0.001	0.553
Waist circumference, cm	0.019	1.020 (1.010–1.030)	<0.001	0.557
Alcohol consumption				
yes (vs. no)	0.212	1.240 (1.080–1.420)	0.002	0.526
Smoking			<0.001	0.653
Past vs. Never	1.212	3.360 (2.860–3.960)	<0.001	
Current vs. Never	1.292	3.640 (3.050–4.350)	<0.001	
Physical activity				
yes (vs. no)	−1.801	0.860 (0.740–0.990)	0.032	0.518
Blood pressure				
Systolic blood pressure, mmHg	0.010	1.010 (1.010–1.010)	<0.001	0.557
Diastolic blood pressure, mmHg	−0.034	0.970 (0.960–0.970)	<0.001	0.590
Diabetes mellitus test				
Fasting serum glucose, mg/dL	0.004	1.004 (1.001–1.006)	0.006	0.544
Dyslipidemia test				
Total cholesterol, mg/dL	−0.007	0.993 (0.991–0.995)	<0.001	0.579
Triglyceride, mg/dL	0.000	1.000 (0.999–1.001)	0.911	0.515
Liver function test				
Aspartate aminotransferase, IU/L	0.004	1.004 (0.998–1.009)	0.176	0.542
Alanine aminotransferase, IU/L	−0.007	0.993 (0.998–0.999)	0.013	0.509
Kidney function test				
Blood urea nitrogen, mg/dL	0.053	1.050 (1.040–1.070)	<0.001	0.574
Creatinine, mg/dL	1.663	5.280 (3.890–7.160)	<0.001	0.641
Urine test				
Urinary protein (vs. Negative)	0.169	1.180 (1.050–1.330)	0.007	0.515
Urinary glucose (vs. Negative)	0.025	1.030 (0.940–1.120)	0.590	0.506
Urinary pH	0.023	1.020 (0.940–1.110)	0.579	0.504

ORs calculated using logistic regression. Acronyms: OR, odds ratio; CI, confidence interval.

**Table 3 jcm-11-03181-t003:** Multivariable model for prediction of COPD in the training cohort.

Covariate	Estimate	OR (95% CI)	*p*-Value
Intercept	−3.582		<0.001
Age, years	0.083	1.087 (1.075–1.095)	<0.001
Sex, female	−1.636	0.195 (0.164–0.230)	<0.001
Waist circumference, cm	−0.019	0.981 (0.972–0.990)	<0.001
Diastolic blood pressure, mmHg	−0.015	0.985 (0.977–0.993)	<0.001

ORs calculated using logistic regression. Acronyms: OR, odds ratio; CI, confidence interval.

**Table 4 jcm-11-03181-t004:** Performance of the COPD score in the validation cohort.

	COPD Score, Range	COPD, n (%)	OR (95% CI)	*p*-Value
1st quartile	−3.872 (−5.435–−3.273)	918 (24.986)	1.000 (reference)	
2nd quartile	−2.852 (−3.272–−2.460)	919 (25.014)	1.838 (1.156–2.923)	0.010
3rd quartile	−1.998 (−2.460–−1.509)	919 (25.014)	4.297 (2.827–6.534)	<0.001
4th quartile	−0.745 (−1.509–−1.026)	918 (24.986)	15.553 (10.484−23.071)	<0.001

ORs calculated using logistic regression. Acronyms: OR, odds ratio; CI, confidence interval.

**Table 5 jcm-11-03181-t005:** Performance of the COPD score for CVD, CHD and stroke.

	Estimate	OR (95% CI)	*p*-Value	C-Index
CVD	0.665	1.945(1.776–2.130)	<0.001	0.730
CHD	0.755	2.128(1.896–2.390)	<0.001	0.758
Stroke	0.507	1.661(1.459–1.890)	<0.001	0.683

ORs calculated using logistic regression. Acronyms: OR, odds ratio; CI, confidence interval; CVD, cardiovascular disease; CHD, coronary heart disease.

## Data Availability

All data used in the present study were obtained from KNHANES, conducted by KCDC, which is open to members of the public and is accessible at https://knhanes.kdca.go.kr/knhanes/sub03/sub03_01.do (accessed on 10 April 2022).

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
