# Peer review of "Health-Screening-Based Chronic Obstructive Pulmonary Disease and Its Effect on Cardiovascular Disease Risk"

_jcm, 2022, doi:10.3390/jcm11113181_

Round 1

Reviewer 1 Report

Thank you for the manuscript entitled "Development and Validation of a Health Screening-based Chronic Obstructive Pulmonary Disease Prediction Model". The study used data from over 10,000 individuals to develop a predictive model for screening for COPD with an area under the curve of the COPD score of 0.7797. It was very nice that the authors included the TRIPOD checklist to further strengthen the manuscript.

Specific comments:

  • There are many studies already published which are using the same idea even with more sophisticated ways of learning representation. All studies are missing in the introduction/literature review.
  • How do you split the dataset into the training and test cohorts? Please specify.
  • Table 1 is difficult to read because you centred the text in the first column. It might be better if you align the text to the left and indent the sub characteristics (Yes/No) a bit.
  • Page 4 Line 134: You forgot the left bracket.
  • Some statistical methods need more explanation.
  • Table 2: What is the unit of urinary protein and urinary  glucose?
  • It is hard to believe that your COPD score only includes age, gender, WC, and DBP. (I expect to see some smoking-related attributes)  I know it comes from statistics. There may be a confounding effect that you need to correct for. Can you provide more evidence on this? Can you say more about patients with CHD/CVD? What is the proportion of CVD/CHD patients in the data set? If you are aiming for your algorithm to work as a screening tool for the generation population. Your dataset needs to be taken from the general population. If you remove patients with CVD/CHD from your training cohort, do you think the statistics would yield a different set of variables?
  • Page 5 Line 157 - 159: Would it be better if you separate the equations from the main text? 
  • I think it is a strong claim when you say that this is the first study to develop a COPD score with only basic variables. The comparison of the developed model with the model proposed by others has not been addressed in detail. Please provide a more direct comparison to further strengthen the manuscript and highlight the research gaps.
  • Various typographical and grammatical errors need to be corrected.
  • It would be nice to see a separate conclusion section.
  • A typographical error in Reference #14 

Author Response

Thank you for your detailed comment. I am attaching a word document for your answers

Reviewer 2 Report

Authors introduced a risk score to evaluate the probability of Chronic obstructive pulmonary disease from the observational study KNHANES. Below are my comments.

P values calculated using t test (continuous variable) and chi-square test (categorical variable). -Have you performed multiple test correction?

The motivation for feature selection was unclear. Including all variables with trained regularization parameter, e.g. elastic net regression would be more practical as the number of variables is fairly small and no overfitting is expected.

Application of C-index from the univariate modelling is not clear, what is the threshold used for feature inclusion? Selecting one feature (WC) over the other (BMI) with the C-index near ~0.55 seems unjust.

Figure S1 should be in the main text, along with the validation dataset model summary – this is equally important to the model performance on the training set.

Why do you estimate univariate results for gender-stratified dataset? What are the known covariates of the COPD? Table 5 shows that this model can predict three other major condition, so the COPD score based on age, sex, waist size and blood pressure is simply predicting an overall poor health status. You need to include more variables to make it more specific to COPD

My other biggest concern is that 949/6088 case/control ratio is a very imbalanced dataset. Using ROC AUC can be misleading, the performance values can be inflated, so authors need to report additional metrics such PRC and/or balanced accuracy to account for modeling imbalanced dataset.

Minor comments:

spirometry test – add some brief info about the meaning of the test

COPD individuals were defi­­­ned as aged 40 years or older with prebronchodilator 89 forced expiratory volume in 1 s/forced vital capacity (FEV1/FVC <0.7) [13]. – You used 20 years threshold in the diagram and 2.1 text, which one is correct?

When smoking is further 150 added as a component of the model, the difference in C-index only increased by 0.1 – you mean by 0.01?

Author Response

(The authors gave the same response as above.)

Round 2

Reviewer 1 Report

The authors have addressed all of my comments and concerns in the revised version.

Author Response

Thank you very much for your constructive and valuable comments, which have helped us improve our manuscript. The manuscript has been carefully rechecked, and appropriate changes have been made following the reviewer’s comments.

Sincerely,

Hyun Wook Han, MD, PhD

Department of Biomedical Informatics, CHA University School of Medicine, CHA University, Seongnam, 13488, Republic of Korea

Tel.: +82-31-881-7109

E-mail: stepano7@gmail.com

Reviewer 2 Report

Figure 2 needs an extended label: A-D subplots description/smoking status

Moreover, significant estimates were made for CVD risks using the COPD score. The ef- fect of the developed COPD prediction score on CVD occurrence prediction was con- firmed, and COPD prediction is possible only with the results of a general examination report. This suggests that there is a potential to predict overall health, such as CVD and CHD.  - this needs to be reflected in the title

Author Response

Thank you very much for your constructive and valuable comments, which have helped us improve our manuscript. The manuscript has been carefully rechecked, and appropriate changes have been made following the reviewer’s comments.
